# The Hidden Cost of High Aspirations: Examining the Stress-Enhancing Effect of Motivational Goals Using Vignette Methodology

**DOI:** 10.3390/ejihpe15070128

**Published:** 2025-07-10

**Authors:** Tamara Gschneidner, Timo Kortsch

**Affiliations:** 1Faculty of Psychology, FernUniversität in Hagen, 58097 Hagen, Germany; tamara.gschneidner@studium.fernuni-hagen.de; 2Department of Social Sciences, IU International University of Applied Sciences, 99084 Erfurt, Germany

**Keywords:** stress, motivational goals, individual differences, occupational stressors

## Abstract

Occupational stress is a major contributor to mental and physical health problems, yet individuals vary in how they appraise and respond to stress, even in identical situations. This study investigates whether motivational goals and internalized conflict schemas—as proposed by Grawe’s Consistency Theory—account for these differences by intensifying subjective stress when approach and avoidance goals are simultaneously activated. In a vignette-based pilot study, we validated 12 workplace scenarios varying in incongruence levels. In the main study (*N* = 482; mean age 25 years; 83.2% female), participants completed the FAMOS questionnaire to assess approach and avoidance goals and were randomly assigned to 4 out of the 12 pretested vignettes. Subjective stress was measured before and after vignette exposure using the SSSQ, and subjective wellbeing was measured using the PANAS. Multilevel modeling showed that participants with stronger avoidance goals and conflict schemas reported higher baseline stress, and that experimentally induced high incongruence led to greater increase in stress levels compared to low incongruence in three out of four scenarios. These findings suggest that psychological inconsistencies—particularly avoidance goals, conflict schemas, and goal incongruence—serve as internal stressors that intensify stress responses. The results highlight the importance of considering individual motivational patterns in stress research and intervention.

## 1. Introduction

Work-related stress is well documented for its detrimental effects on both physical and psychological health, as well as on individuals’ attitudes and behaviors in the workplace ([45]; [10]). Chronic stress can impair physical health, contributing to conditions such as hypertension or weakened immune function ([66]; [29]), and it is also associated with psychological disorders such as depression ([12]). In addition, stress negatively impacts work productivity and the economic success of organizations ([37]). A meta-analysis of factors related to job stress identified key stressors such as workload, role conflict, and role ambiguity ([48]). To mitigate these effects, various stress management interventions have been developed and evaluated. Evidence suggests that such interventions, particularly when implemented at both the individual and organizational level, can be effective in reducing workplace stress and enhancing psychological well-being ([78]; [77]; [67]). Meta-analytic findings further support the efficacy of workplace stress interventions, showing significant improvements in health outcomes and work-related functioning ([7]; [52]).

However, despite these positive effects, it is important to recognize that individuals do not respond to stressors in a uniform manner. As [8] ([8]) emphasize, the stress process comprises two central components: exposure to stressors and individual reactivity. Even when exposed to the same environmental conditions, people can show markedly different emotional and physiological stress responses.

These variations are influenced by a combination of factors, including genetic predispositions, early life experiences, personality traits, cognitive interpretations, and the presence or absence of social support ([71]). For example, an experience of social exclusion—such as being deliberately left out of a group conversation during lunch—may trigger very different responses in different individuals. Individual’s needs ([26]) and associated motivational goals play a central role in shaping these stress responses. A person with a strong need for social connectedness is likely to experience such exclusion as significantly more distressing than someone with a lower need for relatedness. In psychological research, the concept of stress is defined differently depending on the underlying theoretical framework. These individual differences highlight the importance of tailoring stress management interventions to personal needs and psychological profiles. Both the Cybernetic Theory of Stress ([19]) and Grawe’s Consistency Theory ([32], [33]) emphasize that stress arises not only from external demands but also from internal processes—particularly unmet motivational goals and frustrated need satisfaction. However, while existing theories underscore the importance of motivational goals and need satisfaction in the emergence of stress, empirical studies investigating their direct causal role in subjective stress experiences remain scarce. Instead, there are a few indirect indications from various studies that broadly support the assumptions. Research on approach and avoidance goals by [70] ([70]) offers insights into how avoidance goals can undermine well-being and motivation. However, most studies (e.g., [70]; [17]; [76]) focus on externally induced, context-specific conflicts (e.g., workload, role demands), diverging from the emphasis on internal goal congruence. Although these studies incorporate the concept of approach and avoidance goals, they do so from diverse theoretical traditions (such as the BAS/BIS framework, Achievement Goal Theory, or job crafting), and definitions vary considerably across studies. This conceptual inconsistency limits the ability to unify findings under a common motivational framework. Some studies, such as [30] ([30]), do address internal motivational dynamics more directly. Their findings show that goal conflicts among teachers undermine basic psychological needs like autonomy and competence, reducing autonomous motivation and increasing stress. This supports the broader idea that motivational inconsistencies contribute to psychological strain. A small number of studies drawing on Cybernetic Theory (e.g., [27]; [75]) offer related insights. They show that perceived discrepancies between actual and desired work conditions predict well-being more accurately than subjective perceptions alone and highlight self-regulatory mechanisms in coping with stress (e.g., in email overload).

In summary, while contemporary research provides valuable findings that are compatible with certain aspects of the Consistency Theory, there remains a lack of empirical studies that directly test its central theoretical assumption: that the perceived consistency of internal motivational structures is a key determinant of psychological well-being.

Although the relationship between motivational goals and stress has been addressed in previous correlational research (e.g., [28]), a systematic experimental investigation is still lacking. Therefore, the aim of the present study is to address this gap by experimentally testing whether frustrated motivational goals—operationalized through psychologically relevant vignettes—lead to increases in subjective stress. By applying a randomized vignette design with pre- and post-stress assessments, this study provides an important step toward clarifying the internal mechanisms of stress responses and informing more individualized strategies for occupational health and well-being.

## 2. Theoretical Background

[32]’s ([32], [33]) consistency theory, rooted in clinical psychology, offers an empirically based framework linking need satisfaction, motivational goals, and stress. According to this theory, psychological processes are primarily driven by the pursuit of consistency, which refers to the alignment between an individual’s goals, perceptions, and evaluations. Disruptions in this consistency can signal internal conflict, leading to psychological strain ([32]). Building on Epstein’s theory ([26]), [33] ([33]) incorporated these basic psychological needs into his consistency theory, positing that individuals are fundamentally motivated by the need to satisfy four core needs: orientation and control, pleasure attainment and pain avoidance, self-esteem enhancement and protection, and attachment and affiliation. To achieve these psychological needs, motivational schemas are formed, guiding behavior and influencing how individuals respond to challenges. These motivational schemas involve the coordination of simultaneously active motivational, cognitive, and neural processes in a coherent manner. Ideally, these processes should be integrated with minimal internal conflict to facilitate effective coping and adaptation to the environment. In other words, the more successful the coordination of interacting motivational, cognitive, and neural processes, the better individuals can navigate their environment and maintain psychological well-being ([33]).

### 2.1. Motivational Schemas and Goals

According to [32] ([32]), individuals possess an inherent drive for need satisfaction, which leads to the development of motivational schemas aimed at fulfilling these needs. These schemas are complex, hierarchically organized cognitive structures that guide individuals in seeking or avoiding certain perceptions. Motivational schemas not only include individual goals but also the means to achieve them, integrating various goals and associated tendencies to attend to specific information and initiate related actions ([58]; [64]; [32]; [62]). Essentially, they are mental representations that help coordinate cognitive and behavioral responses to achieve or avoid particular outcomes.

[32] ([32]) identifies three key types of motivational schemas used in psychotherapy: intentional schemas, avoidance schemas, and conflict schemas. Intentional schemas are related to approach goals, which drive individuals to actively pursue and achieve desired states. In contrast, avoidance schemas are connected to avoidance goals, aiming to prevent threats or losses. This distinction is grounded in self-regulation theory, as proposed by [11] ([11]), which suggests that behavior is motivated by two distinct types of feedback processes. One type of behavior seeks to reduce existing discrepancies between the current state and a desired goal, reflecting an approach-oriented motivation. The other type is driven by the desire to avoid advancing towards anti-goals, thus maintaining or increasing discrepancies intentionally. In this way, approach goals are directed toward satisfying basic psychological needs, while avoidance goals aim to prevent negative outcomes. Both types of goals are categorized as process goals, meaning they represent an ongoing effort to approach or avoid specific states, rather than being ultimately achievable. According to [32]’s ([32]) consistency theory, all individuals possess both systems, but the degree to which each is emphasized may vary depending on individual socialization experiences (see also [13]). Additionally, conflict schemas represent situations where approach and avoidance goals are simultaneously activated, leading to interference in the execution of each goal. This simultaneous activation can create internal conflict, making it challenging to effectively pursue either goal, thereby contributing to psychological strain.

### 2.2. Stress and Well-Being

According to [32] ([32], [33]), psychological inconsistency arises when an individual’s psychological processes are not aligned with their basic needs, resulting in conflicts that hinder goal achievement and contribute to stress. Psychological inconsistency is defined as the incompatibility that occurs when conflicting motivational processes are activated simultaneously, creating a state of internal conflict. [32] ([32]) emphasized that psychological inconsistency, which manifests as a discrepancy between internal needs and external reality, is often accompanied by the frustration of fundamental psychological needs, such as autonomy, control, attachment, and self-esteem. In this framework, inconsistency can be further categorized into discordance and incongruence.

#### 2.2.1. Avoidance Goals, Conflict Schemas, and Discordance

Individuals with a strong focus on avoidance goals tend to expect negative outcomes, which is associated with lower subjective well-being ([13]; [23]; [24]). This avoidance orientation, if overly emphasized, acts as a barrier to achieving approach goals and is linked to high inconsistency and should therefore lead to higher subjective stress. Avoidance goals are considered particularly problematic, as they are more likely to generate internal conflicts (discordance) and substantially increase the risk of experiencing incongruence ([34]). These considerations form the basis for the following hypothesis H1:
**H1.** *Higher levels of avoidance goals lead to (a) higher levels of stress and (b) lower well-being.*

If avoidance goals gain excessive importance due to numerous harmful experiences, approach goals can only be achieved if avoidance goals are simultaneously violated. In this case, motivational tendencies are considered incompatible ([28]). Discordance refers to the incompatibility of goals and arises when conflicting approach and avoidance schemas are activated, also known as conflict schemas. This discordance or conflict in motivated psychological processes leads to less effective goal achievement compared to when there is a clear focus of psychological activity. It results in incongruence between the conflicting goals, which increases the level of psychological inconsistency. According to a meta-analysis by [28] ([28]), discordance (conflict schemas) is positively associated with negative affect, anxiety, and depression, and it is negatively associated with positive affect. The hypothesis H2 based on this theoretical framework is as follows:
**H2.** *Higher levels of conflict schemas lead to (a) higher levels of stress and (b) lower well-being.*

#### 2.2.2. Incongruence

Incongruence is a type of inconsistency, derived from [64]’ ([64]) control theory, referring to the perception that goal achievement does not align with activated goals. According to [32] ([32]), motivational incongruence specifically addresses the role of aversive reference signals (avoidance goals). Since incongruence always involves the thwarting, blocking, or failure of goals, it is associated with negative emotions ([53]). Prolonged incongruence regarding important motivational goals can lead to aversive arousal ([28]) and is considered a significant form of stress, with detrimental effects on the neural, hormonal, and immune systems ([43]).

Motivational incongruence shows strong correlations with symptom distress, depressive mood, reduced life satisfaction, and neuroticism, underscoring its central role in psychological well-being ([34]). The Cybernetic Theory of Stress ([19]) supports this view. It is based on a feedback system focused on self-regulation, where stress, coping, and well-being are part of a negative feedback loop aiming to reduce discrepancies between external influences and internal standards. According to this theory, stress arises from a discrepancy between a person’s perceptions and desires, which must be significant for the person. Perceptions are subjective representations of events and conditions, relating to the past, present, or future. Desires include conscious goals, values, and interests, referred process goals by [64] ([64]). The significance of discrepancy is linked to how central it is to a person’s well-being. Studies have shown that increased discrepancies between reality and personal desires/goals correlate with decreased well-being ([20], [21]; [22]). Both Consistency Theory ([32], [33]) and the Cybernetic Theory of Stress ([19]) suggest that experimentally induced incongruence leads to increased subjective stress. Based on these findings, the following hypothesis H3 is derived:
**H3.** *Experimentally induced high incongruence leads to greater increase in stress levels compared to low incongruence.*

## 3. Materials and Methods

### 3.1. Study Design

This study employed an experimental vignette design (factorial survey; [69]) to examine the effects of avoidance goals, conflict schemas, and incongruence on subjective stress levels and well-being. Vignette studies combine the features of traditional experimental designs and questionnaire studies to balance the strengths and weaknesses of both approaches ([5]). Vignettes allow for testing causal hypotheses ([1]). Vignettes are brief, constructed descriptions of a person, object, or situation that represent a systematic combination of characteristics ([74]). This design was used in this study to manipulate incongruence (the discrepancy between goals and reality) across four distinct situations. To examine whether greater incongruence truly leads to increased stress and reduced well-being, the vignette situations were thematically developed using four motivational schemas from subscales of the FAMOS questionnaire (Failure, Devaluation, Dependence, and Loss of Control), with three different levels of incongruence (small, medium, large) for each category. For each of the four schemas, three work-related scenarios were generated, each reflecting different levels of goal-reality discrepancy. Since a separate analysis was conducted for each of the four subscales to test the hypotheses, each model results in a 2 × 3 (2 time of measurements, 3 groups) design.

### 3.2. Pilot Study

To validate that the vignettes were correctly associated with the right motivational schema (Failure, Devaluation, Dependence, Loss of Control) and the correct level of incongruence (small, medium, large), a pilot study was conducted. The pilot study, conducted in November 2024, aimed to validate the vignettes. The participants were first provided with definitions of each motivational schema. They were then asked to assign the vignettes to the corresponding motivational schema using a drag-and-drop method. The results showed that the participants (*N* = 148) were able to correctly match the vignettes to the motivational schemas in 70–81% of the cases, with chi-square goodness-of-fit tests (χ^2^(4) = 240.19–348.51, *p* < 0.001) showing significant differences between the expected and observed frequencies. Furthermore, when asked to assign the situations to the appropriate incongruence category (small, medium, large), the participants (*N* = 224) correctly categorized the vignettes in 67–82% of the cases. Chi-square tests (χ^2^(2) = 122.82–244, *p* < 0.001) also confirmed significant differences. These results provide evidence for the validity of the vignettes, as they aligned well with the participants’ conceptual understanding. Therefore, the vignettes were adopted for the main study to test the hypotheses. Table 1 contains the participant numbers for each experimental group and detailed pilot study results validating the vignettes, while Appendix A contains the translated vignettes used in the main study.

### 3.3. Study Participants

The participants were recruited via a link sent to students in distance learning programs at the International University of Applied Sciences (IU) and via social media channels. Students in distance learning programs are often part-time students, balancing their studies alongside full-time jobs. Therefore, these students typically have significant work experience. Furthermore, work experience was an inclusion criterium for this study: To participate, the respondents had to be at least 18 years old, not undergoing psychological treatment, and have had work experience. A total of 531 participants started the survey between 25 January 2025 and 1 February 2025. The questionnaire was accessed 586 times, with 475 participants completing it in full. Participants who demonstrated inattentive or dishonest responses were excluded, checked via manipulation checks. Additionally, 10 participants were excluded due to one-sided response patterns. After filtering, a total of 482 valid datasets remained. The participants ranged in age from 16 to 63 years (M = 25.29, SD = 6.97). The majority were female (83.2%) and had higher education (80.5%). A total of 336 participants were students, and 71% of the participants were employed in part-time or full-time jobs. A total of 86.31% of participants lived in Germany. Additionally, 29 participants had previously participated in the pilot study.

### 3.4. Data Collection Procedure and Data Collection Tools

The data was collected online using the survey software SoSci Survey (version 3.7.00; [54]), a widely used and established survey platform in academic research in Germany. SoSci Survey is known for its accessible survey interface design and compliance with strict German data privacy regulations, which was particularly important given the vulnerability of the student sample. Before data collection, the participants were informed of the objectives and methods of the study, including their right to decline participation. An informed written consent form was signed by each participant. Anonymity in data collection was ensured by not collecting information that could reveal the participants’ identity. The participants first completed a series of questionnaires, including the Motivational schema Analysis Questionnaire (FAMOS) ([35]). The other questionnaires were not included in this paper. After completing these questionnaires, the participants began the vignette experiment. They first reported their current stress level (using the Short Stress State Questionnaire in German (SSSQ-G); Ringgold et al., 2024) and current mood (measured with the Positive and Negative Affect Schedule (PANAS); Krohne et al., 1996). The participants were then asked to read and mentally immerse themselves in the presented scenarios. Each participant was randomly assigned one of the three incongruence levels for each vignette. The vignettes were presented in the following fixed order: Failure, Devaluation, Dependence, and Loss of Control. After each vignette, the participants were asked to indicate their stress levels and mood once more. Finally, demographic questions were posed, along with questions about their guesses regarding the study’s background and previous participation in the pilot study. The online survey took approximately 45 min to complete. Below, the instruments used in the present study are described. The participants were asked about their age, gender (gender identity), educational background, occupation, and current place of residence. They were also asked if they had previously participated in the pilot study.

### 3.5. Instruments

Avoidance goals. Avoidance goals were assessed using the validated Motivational Schema Analysis Questionnaire (FAMOS; [35]). In line with Grawe’s Consistency Theory ([32], [33]) and the four developed vignettes, avoidance goals were operationalized through four specific subscales of the FAMOS. The FAMOS includes 14 subscales for approach goals and 9 for avoidance goals. For this study, four pre-existing avoidance goal subscales were used: Failure, Devaluation, Dependence, and Loss of Control. Each subscale consisted of 4–5 items, rated on a 5-point Likert scale ranging from 1 (“not important (bad)”) to 5 (“extremely important (bad)”). The internal consistency of the avoidance goal scales was satisfactory, with McDonald’s Omega ω = 0.78.

Conflict schemas. To operationalize the simultaneous activation of avoidance and approach goals, four corresponding subscale pairs from the Motivational Schema Analysis Questionnaire (FAMOS; [35]) were used: Performance and Failure, Recognition and Devaluation, Independence and Dependence, and Control and Loss of Control. For each domain, the sum of the scores of the respective avoidance and approach subscales was calculated and then multiplied to represent the degree of simultaneous activation. For example, for the domain Performance and Failure: Sum of the Failure subscale × Sum of the Recognition subscale. High values in both scales result in high values in the conflict schema. McDonald’s Omega for approach goals was ω = 0.80, and for avoidance goals, it was ω = 0.78, indicating satisfactory internal consistency.

Perceived stress. Perceived stress was measured using the Short Stress State Questionnaire in German (SSSQ-G; [68]), a German version of the Short Stress State Questionnaire by [38] ([38]). Both pre- and post-stressor versions of the questionnaire were used, with identical items but adjusted time references (e.g., “I want to succeed in the task” vs. “I wanted to succeed in the task”). The responses were measured on a 6-point Likert scale (1 = “not at all” to 6 = “extremely”). The SSSQ-G consists of 24 items, assessing subjective emotional, motivational, and cognitive states related to stress. Higher scores indicate higher perceived stress. In this study, McDonald’s Omega values indicated good to excellent internal consistency, with ω = 0.90 for the pre-stressor version and ω = 0.93 for the post-stressor version (corresponding Cronbach’s alpha values of α = 0.87 and α = 0.89, respectively).

Wellbeing. According to [80] ([80]), individuals are considered happy when they experience frequent positive and rare negative emotions. The Positive and Negative Affect Schedule (PANAS; [51]) was therefore used to assess subjective well-being. PANAS consists of 20 items rated on a 1–5 Likert scale (1 = “not at all” to 5 = “extremely”), with 10 items assessing positive affect (PA) and 10 items assessing negative affect (NA). Scores for positive and negative affect are averaged to create composite scores for each. In this study, the PANAS scales demonstrated good to excellent internal consistency. For the pre-assessment, Cronbach’s alpha was α = 0.90 and McDonald’s Omega was ω = 0.92 for both the PA and NA scales. For the post-assessment, Cronbach’s alpha was α = 0.93 (ω = 0.95) for the PA scale and α = 0.94 (ω = 0.95) for the NA scale. Only the PANAS-NA subscale was analyzed for this study, with higher values indicating lower well-being.

### 3.6. Preparation of the Dataset and Data Analysis

Prior to the analysis, the dataset was preprocessed, extracting only the scales necessary for the analyses, some of which contained missing values (3.4% in total). Due to the small amount of missing data, a [55] ([55]) MCAR test could not be performed. Logistic regressions indicated no significant results, suggesting that the missing data did not systematically influence the outcomes. Scale values were calculated by summing or averaging the item values, with missing data handled according to [31] ([31]) guidelines. If more than half of the variables were available for scale creation and item–total correlations were consistent, scales were computed. However, for the SSSQ scales and the pre-PANAS measurement, the domain-sampling model ([61]) was not confirmed due to varying item–total correlations; thus, these scales were marked as “NA.” Four separate datasets were created for each motivational schema to test the models. Missing values after scale creation ranged from 1.1% to 2.3%. The data were originally in wide format, with one row per participant. This was converted to long format. Due to the minimal missing data, cases with missing values were listwise deleted, as this did not pose a major threat to statistical power.

Descriptive and inferential analyses were performed using R (Version 4.0.5; [65]) with RStudio (Version 2024.09.1; [63]) and JASP (Version 0.19.3.0; [44]). Given the hierarchical structure of the data, hierarchical multilevel models were applied, which are recommended for vignette designs ([2]). While ANCOVA could also be used, multilevel analysis offers key advantages, such as modeling individual development trajectories, handling correlated measurements within participants, and accommodating missing data, which can lead to more precise estimates and higher statistical power ([42]).

For each of the four motivational schemas, two separate models were calculated for the two dependent variables (stress perception and well-being), with the hierarchical structure defined by two measurement points (Level 1) nested within the participants (Level 2). Each model was used to test all three hypotheses for the respective motivational schema. The following model labels were applied: Model Failure (Stress/Wellbeing), Model Devaluation (Stress/Wellbeing), Model Dependence (Stress/Wellbeing), and Model Loss of Control (Stress/Wellbeing). The analysis of Hypothesis H1 was conducted using the following avoidance goals: Failure, Devaluation, Dependence, and Loss of Control. For Hypothesis H2, the analysis focused on the following conflict schemas: Performance vs. Failure, Recognition vs. Devaluation, Independence vs. Dependence, and Control vs. Loss of Control. H3 was tested using the four incongruence vignettes for Failure, Devaluation, Dependence, and Loss of Control. Predictor variables included group membership (dummy-coded), measurement timepoint, and the time-invariant variables for avoidance goals and conflict schemas. Grand Mean Centering ([25]) was used for centering the continuous variables of the FAMOS questionnaires (avoidance goals and conflict schemas). Four hypotheses were tested by examining main effects (H1, H2, H3, H4), and the fifth hypotheses was tested by examining cross-level interaction effects as well as simple slope analysis for interaction effects.

### 3.7. Model Estimation

The model parameters were estimated using the maximum likelihood method. Treatment coding (dummy coding) was applied, with Group 1 (no to small incongruence) defined as the reference category, or control group. The experimental groups (Groups 2 and 3) were interpreted relative to the control group, as the interest lies in how the groups with moderate and high discrepancy changed over time compared to the control group. Each parameter was added to the model sequentially. All four interaction models for the dependent variable stress showed a significantly better model fit compared to their respective null models, as indicated by the likelihood ratio test (see Table 2). Except for the “Dependence” model, all four models for the dependent variable well-being also demonstrated a significantly improved model fit compared to their corresponding null models (see Table 3). The intraclass correlations (ρ = 0.211 to 0.447) suggested that part of the total variance could be explained by interindividual differences, supporting the hierarchical structure. The results are presented in Table 2 and Table 3. More detailed information on the model estimation and the assessment of the model fit (Table 2 and Table 3) is provided in Appendix B.

## 4. Results

### 4.1. Assumptions and Descriptive Analyisis

For each dataset, the assumptions for multilevel analysis ([42]) were tested, as the model results can only be interpreted if the assumptions are met. The assumptions of normal distribution, homoscedasticity, absence of residual outliers, linearity, and absence of multicollinearity were generally satisfied. The assumption of homoscedasticity was assessed via scatterplots. For the dependent variable stress, the models “Failure,” “Devaluation,” and “Dependence” showed no systematic patterns, suggesting the assumption was met. However, for the model “Loss of Control,” and all models predicting well-being, heteroscedasticity was observed. To address this, models were re-estimated using cluster-robust standard errors (CR2), clustering by participant ID to account for within-subject dependencies. Outliers were assessed using boxplots; although some positive residuals were detected, they appeared substantively plausible and were retained. For *well-being* models, a sensitivity analysis excluding observations with standardized residuals > |2.5| confirmed the robustness of the main effects. The means and standard deviations of the measured variables are summarized in Table 4 and Table 5.

### 4.2. Inferential Hypothesis Testing

#### 4.2.1. Failure, Stress, and Wellbeing

For the Model Failure (stress/wellbeing), hypotheses H1 and H2 were supported. Higher levels of avoidance goals and conflict schemas lead to significantly higher levels of stress and lower well-being. The results of the linear mixed model analysis are detailed in Table 6. The main effects of the Level 2 predictors for the avoidance goal and conflict schema were significant and thus had a direct impact on both the subjective stress level (SSSQ_total) and subjective well-being (PANAS_n). H3 was also supported for the Model Failure. Experimentally induced high incongruence led to a greater increase in stress levels compared to low incongruence (Figure 1). The cross-level interactions (time of measurement×group membership for groups 2 and 3) reached statistical significance. Pairwise comparisons revealed no significant change in stress levels within the control group (low incongruence) between the time of measurement 0 (M_0_ = 56.7) and the time of measurement 1 (M_1_ = 55.0), t(452) = 1.49, *p* = 0.67. However, in the first experimental group (moderate incongruence), a significant increase in stress levels was observed from the time of measurement 0 (M_0_ = 56.1) to the time of measurement 1 (M_1_ = 68.2), t(452) = −11.02, *p* < 0.001. The second experimental group (high incongruence) also showed a significant increase in stress, from M_0_ = 58.4 to M_1_ = 76.0, t(452) = −15.80, *p* < 0.001. These results indicate that the stimulus for the motivational schema “Failure” led to a significant increase in subjective stress in both experimental groups, while no change was observed in the control group.

#### 4.2.2. Devaluation, Stress, and Wellbeing

Hypotheses H1 and H2 were supported. For Devaluation, higher levels of avoidance goals and conflict schemas led to significantly higher levels of stress and lower well-being. The results of the linear mixed model analysis are detailed in Table 7. The mean value of the avoidance goal Devaluation had a positive effect on SSSQ_total, with a one-unit increase being associated with a 4.16-point increase in SSSQ_total (b = 4.16, t(462) = 2.88, *p* = 0.004). Similar results were found for the conflict schema (b = 0.73, t(462) = 3.18, *p* = 0.002) and for the dependent variable well-being (see Table 7). H3 was also supported for the Model Devaluation. Experimentally induced high incongruence led to greater increase in stress levels compared to low incongruence (Figure 2). The interaction between time of measurement and group membership was also significant. Pairwise comparisons revealed that in Group 1 (low incongruence), stress significantly decreased (M_0_ = 56.4, M_1_ = 45.8; t(452) = 9.32, *p* < 0.001). In Group 2 (Experimental Group 1), stress significantly increased (M_0_ = 56.2, M_1_ = 71.5; t(452) = −10.11, *p* < 0.001). Similarly, in Group 3 (Experimental Group 2), a significant increase in stress was observed (M_0_ = 58.1, M_1_ = 79.5; t(452) = −15.75, *p* < 0.001). These findings suggest that the stimulus led to an increase in subjective stress in both experimental groups, whereas a significant reduction in stress was observed in the control group.

#### 4.2.3. Dependence, Stress, and Wellbeing

Hypotheses H1 and H2 were not supported. For Dependence, higher levels of avoidance goals and conflict schemas did not lead to significantly higher levels of stress and lower well-being. The results of the linear mixed model analysis are detailed in Table 8. The main effects of the Level 2 predictors for the avoidance goal and conflict schema were not significant and thus had no direct effect on both subjective stress level (SSSQ_total) and subjective well-being (PANAS_neg). H3 was supported for the Model Dependence. Experimentally induced high incongruence led to a greater increase in stress levels compared to low incongruence (Figure 3). The cross-level interactions (measurement time * group membership for Groups 2 and 3) showed significantly higher values compared to the control Group 1. Pairwise comparisons were conducted. In the control group (Group 1), a significant decrease in stress levels was observed from time of measurement 0 (M = 56.8) to time of measurement 1 (M = 50.5), t(452) = 5.60, *p* < 0.001. In the first experimental group (Group 2), however, no significant change was found (M_0_ = 56.4, M_1_ = 54.1), t(452) = 1.99, *p* = 0.348. Similarly, in the second experimental group (Group 3), no significant difference was observed (M_0_ = 57.4, M_1_ = 59.7), t(452) = −1.97, *p* = 0.361. These results suggest that the stimulus for the motivational schema “Dependence” did not have a significant effect on subjective stress levels in the experimental groups. However, a significant reduction in stress was observed in the control group.

#### 4.2.4. Loss of Control, Stress, and Wellbeing

Hypotheses H1 and H2 were supported. For Loss of Control, higher levels of avoidance goals and conflict schemas lead to significantly higher levels of stress and lower well-being. The results of the linear mixed model analysis are detailed Table 9. The main effects of the Level 2 predictors for the avoidance goal and conflict schema were significant and thus had a direct effect on both subjective stress levels (SSSQ_total) and subjective well-being (PANAS_neg). H3 was supported for the Model Loss of Control. Experimentally induced high incongruence led to a greater increase in stress levels compared to low incongruence (Figure 4). The cross-level interactions (measurement time * group membership for groups 2 and 3) reached the level of significance. Pairwise comparisons revealed a significant decrease in stress levels in the control group (Group 1) from the time of measurement 0 (M_0_ = 57.0) to the time of measurement 1 (M_1_ = 50.3), t(452) = 5.9, *p* < 0.001. In the first experimental group (Group 2), stress significantly increased (M_0_ = 57.7, M_1_ = 68.4), t(452) = −9.04, *p* < 0.001. Similarly, in the second experimental group (Group 3), there was a significant increase in stress (M_0_ = 55.7, M_1_ = 71.9), t(452) = −14.12, *p* < 0.001.

These results suggest that the stimulus for the driver “Maintain control” led to a significant increase in subjective stress in both experimental groups, while a significant reduction in stress was observed in the control group.

## 5. Discussion

To understand and manage work-related stress more effectively, it is essential to consider not only external stressors but also internal psychological dynamics. This study aimed to investigate whether motivational processes act as a psychological stressor by employing an experimental vignette design based on Grawe’s Consistency Theory ([32], [33]) and the Cybernetic Theory of Stress ([19]). Because prior research has only provided correlational evidence linking motivational goal frustration to elevated stress levels ([28]), the present study sought to explore this relationship using an experimental vignette design to causally test whether motivational incongruence increases subjective stress experience. The results provide empirical support for the main hypotheses. Higher levels of avoidance goals and conflict schemas were associated with increased stress and decreased well-being (H1–H4), except for the “Dependence” condition. In line with Hypothesis H5, experimentally induced high motivational incongruence led to a significantly greater increase in perceived stress compared to low incongruence in all four scenarios. Multilevel analysis revealed that the participants in the high-incongruence (Group 3) and medium-incongruence (Group 2) conditions experienced a greater increase in stress than those in the control group (Group 1), thereby supporting the assumption that motivational incongruence contributes to psychological strain. Unexpectedly, no significant increase in stress was observed between Time 0 (pre-induction) and Time 1 (post-induction) in any group for the ‘Dependence’ vignette condition. Even more surprisingly, the control group (Group 1)—originally intended as a neutral condition—showed a significant reduction in perceived stress over time in all scenarios except for the ‘Failure’ vignette.

These findings align well with Grawe’s Consistency Theory, which posits that psychological strain results from inconsistencies between motivational goals and perceived reality. In this framework, discordance (conflict schemas) and incongruence (discrepancy between goals and external conditions) are seen as distinct but interacting forms of inconsistency. Situations inducing incongruence led to higher stress levels, thereby supporting both Grawe’s Consistency Theory and the Cybernetic Theory of Stress ([19]), which also emphasizes perceived discrepancy as a core mechanism of stress. Importantly, the control condition, initially expected to produce neutral results, instead led to stress reduction in three out of four vignettes. This suggests that low incongruence may not only prevent stress but also actively promote psychological regulation. Such an effect supports previous findings linking the resolution of motivational incongruence in psychotherapy to improvements in psychological symptoms ([6]; [34]).

The findings in this study furthermore align with prior research showing that motivational inconsistency—such as goal conflicts and avoidance goals—is linked to increased psychological symptoms like stress, anxiety, and depression ([28]). This study adds experimental evidence that even situationally induced motivational incongruence causes acute stress responses, suggesting a causal role. Supporting this, [36] ([36]) found that individuals with high goal conflict benefit more from clarification-focused therapy. Similarly, [30] ([30]) demonstrated that goal conflicts undermine basic psychological needs at work, increasing stress. In addition, [27] ([27]) found that discrepancies between employees’ desired and actual work conditions (so-called GAPs) more strongly predicted well-being than absolute perceptions of the work environment, emphasizing the importance of perceived congruence. Together, these results highlight motivational consistency as crucial for psychological well-being and suggest that addressing motivational conflicts may reduce psychological strain across populations.

The absence of effects in the “Dependence” condition deviated from theoretical expectations. Descriptive results showed high FAMOS Dependence scores, suggesting the participants generally perceived dependence negatively. However, the vignettes in this condition, especially under medium and high incongruence, included elements of successful collaboration and social support. This emphasis on social support may have reduced the perceived stressfulness of the situations. As [39] ([39], [40]) argues, access to social resources is essential for psychological functioning. In occupational settings, support from colleagues and supervisors represents a central resource. These resources can directly enhance outcomes such as job attitudes and social interaction quality (e.g., [41]; [50]), or they can act as buffers against environmental stressors ([79]). From this perspective, the absence of expected stress responses in the presence of incongruence may be attributed to the stress-reducing role of social support described in the vignettes. The Job Demand Control (Support) model ([46]; [47]) and empirical findings ([72]) further support this interpretation, suggesting that job resources such as social support can mitigate the impact of demands on stress. In line with this, the participants may have perceived the supportive aspects of the scenarios as valuable resources, thereby diminishing the stress-inducing potential of the incongruent conditions.

### 5.1. Theoretical Implications

This study has several theoretical implications. First, the findings provide experimental support for the theoretical assumption that psychological inconsistency—manifested through high avoidance goals, conflict schemas, and motivational incongruence—contributes to elevated levels of subjective stress ([33]). While previous research has mostly relied on correlational designs (e.g., [28]), the current study is among the first to experimentally validate Grawe’s Consistency Theory by inducing motivational incongruence under controlled conditions. Second, the results differentiate between two forms of inconsistency—discordance and incongruence—demonstrating that both can function as potent psychological stressors. This finding offers empirical support for Grawe’s theoretical distinction between them. Third, the results align with the Cybernetic Theory of Stress ([19]), which posits that stress arises from discrepancies between personal goals and perceived reality. Cybernetic Theory suggests that stress can be reduced through various coping strategies, including altering the environment, adjusting personal goals, or changing the perceived importance or duration of the discrepancy ([18]). These pathways emphasize the need for individualized interventions that align with personal values and situational demands. By using a vignette-based design to simulate varying levels of discrepancy, the present study provides experimental support for the theory’s central mechanism and demonstrates that even imagined alignment between perceptions and desires can effectively reduce stress. This not only reinforces the existing theory but also extends it by showing that cognitive engagement with hypothetical low-discrepancy scenarios can produce measurable effects on stress appraisal. Fourth, the control condition, designed to remain neutral, produced an unexpected yet theoretically meaningful effect: participants exposed to low-incongruence vignettes exhibited a significant reduction in stress levels. This suggests that the absence of motivational conflict does not merely prevent stress escalation—it may also actively foster stress reduction. This finding resonates with earlier therapeutic studies showing that reducing incongruence leads to clinical improvement ([6]; [34]) and opens new avenues for investigating how mentally simulating congruent situations might have a regulatory effect on emotional states. Previous research further supports the stress-reducing effects of visualization techniques, indicating their promise as an accessible and adaptable tool for stress regulation. ([3]; [60]). A central element of these interventions is the intentional focus on positive, self-chosen mental images that are associated with safety and a sense of control. Studies show that GI can affect perception so strongly that mental imagery may function similarly to real experiences ([49]; [9]). The findings on the effectiveness of Guided Imagery (GI) can be promisingly applied to managing avoidance goals and conflict schemas in high-stress situations. Future research should investigate whether GI interventions targeting low-discrepancy situations can similarly reduce stress in real-world occupational settings.

### 5.2. Practical Implications

From a practical standpoint, there are also several implications. First, these findings emphasize the need for more individualized approaches to stress management that take internal motivational schemas into account. Even though many studies show the effectiveness of stress management interventions in general ([7]; [52]; [67]), there is also much unexplained variance. The validated FAMOS questionnaire ([35]) can serve as a diagnostic tool to identify maladaptive patterns—such as high avoidance goals or persistent goal conflicts—that are likely to contribute to psychological strain. In workplace contexts, occupational psychologists, company health management professionals, or trained coaches could use FAMOS profiles to better understand employees’ motivational structures and adapt stress prevention programs accordingly.

Second, the findings suggest that socially supportive elements can buffer stress responses, even in situations where individuals experience incongruence. Practical implications include fostering peer support networks, implementing mentoring programs, and developing role-playing or narrative-based interventions aimed at strengthening perceived social resources. Such measures could help individuals better cope with stressful or conflicting situations, enhance resilience in organizational settings, and reduce vulnerability to stress-related outcomes.

Third, Guided Imagery (GI) could be leveraged to address motivational schemas, particularly in managing identified avoidance goals or conflict schemas, within the framework of stress management interventions or leadership development programs. In Stress Management Programs, GI can help employees address avoidance goals and conflict schemas by visualizing calm, controlled environments and successfully navigating challenging situations. To further enhance the effectiveness of GI in the workplace, companies could consider integrating it through mobile apps or online platforms, allowing employees to engage in guided imagery exercises tailored to specific scenarios such as “relaxation before a presentation” or “calming after a stressful meeting.” Additionally, companies could create designated relaxation rooms equipped with audio or video-guided imagery sessions, offering employees a dedicated space for short breaks during the day. Incorporating these practices during lunch breaks could also provide a valuable opportunity for employees to recharge and relieve stress. However, while GI is a novel suggestion for stress management, its feasibility in workplace settings needs further elaboration. Practical considerations, such as the work environment, employee engagement, and potential barriers to implementation, should be explored to determine how effectively GI can be incorporated into workplace stress interventions.

### 5.3. Limitations

It is important to acknowledge that this study has certain limitations. The sample consisted predominantly of young female university students (mean age 25 years), and students may have different experiences with occupational stress compared to full-time employees. This may limit the generalizability of the findings. However, the participants ranged in age from 16 to 63 years, and 71% were employed either part-time or full-time. While the student population is overrepresented, the inclusion of employed individuals adds diversity to the sample. For future studies, it is recommended to recruit a broader sample of employees and include a more balanced representation of students to improve the generalizability of the findings. Furthermore, cultural and social differences ([4]; [73]; [16]; [57]) may influence perceptions of stress; therefore, future research should also include more diverse samples from universities in different countries to enhance the external validity and generalizability of the results. In addition, the reliance on self-reported data for assessing stress may introduce common method bias. Although our pre–post experimental design and randomization strengthen causal inference and mitigate endogeneity concerns, self-reports remain susceptible to shared method variance. Moreover, the simultaneous measurement of variables within one session may lead to inflated associations. To address this, future studies should consider using multiple time points—e.g., measuring certain constructs well before or after the intervention—to disentangle temporal effects more clearly. Furthermore, incorporating physiological indicators (e.g., cortisol) or observer-rated assessments would provide more robust evidence and increase validity. Control variables such as personality traits or coping styles were not included, which may have influenced outcomes. Although manipulation checks revealed no systematic bias, the calming effect in the control condition may point to an unintentional regulatory function of the vignette format itself.

### 5.4. Conclusions

The current study offers experimental evidence supporting the idea that psychological inconsistency—stemming from avoidance goals, conflict schemas, and motivational incongruence—leads to increased stress. In contrast, mentally simulating congruent scenarios can actively reduce this stress. These findings emphasize that stress responses are significantly shaped by individual factors, particularly one’s internal motivational structures. As such, this study highlights the need for more personalized stress interventions that are tailored to align with the specific motivational profiles of individuals, ultimately fostering more effective and targeted stress management strategies.

## Figures and Tables

**Figure 1 ejihpe-15-00128-f001:**
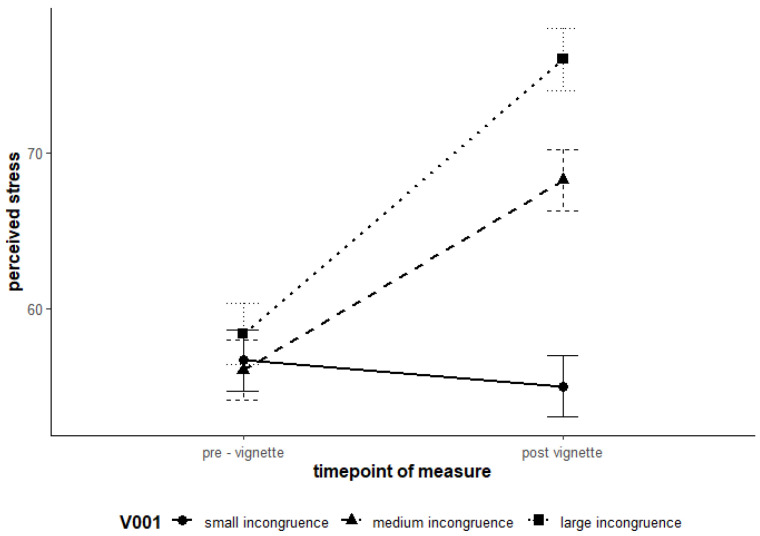
Interaction effects of time and the three incongruence groups on stress regarding the “Failure” vignettes. Note. Error bars represent 95% confidence intervals.

**Figure 2 ejihpe-15-00128-f002:**
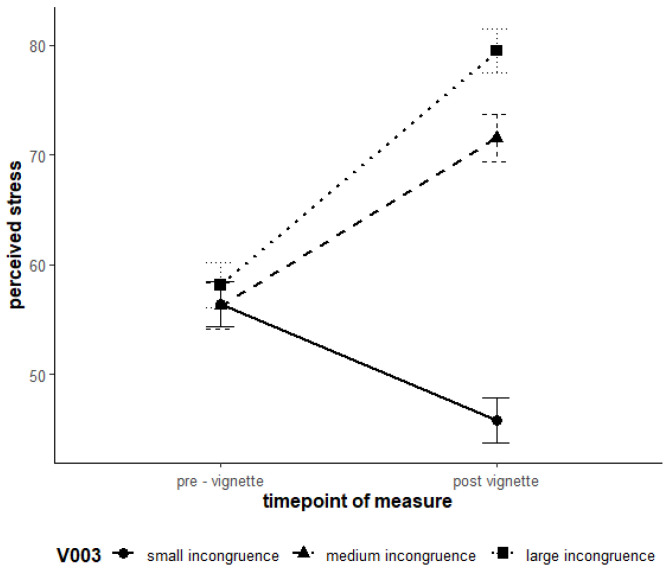
Interaction effects of time and the three incongruence groups on stress regarding the “Devaluation” vignettes. Note. Error bars represent 95% confidence intervals.

**Figure 3 ejihpe-15-00128-f003:**
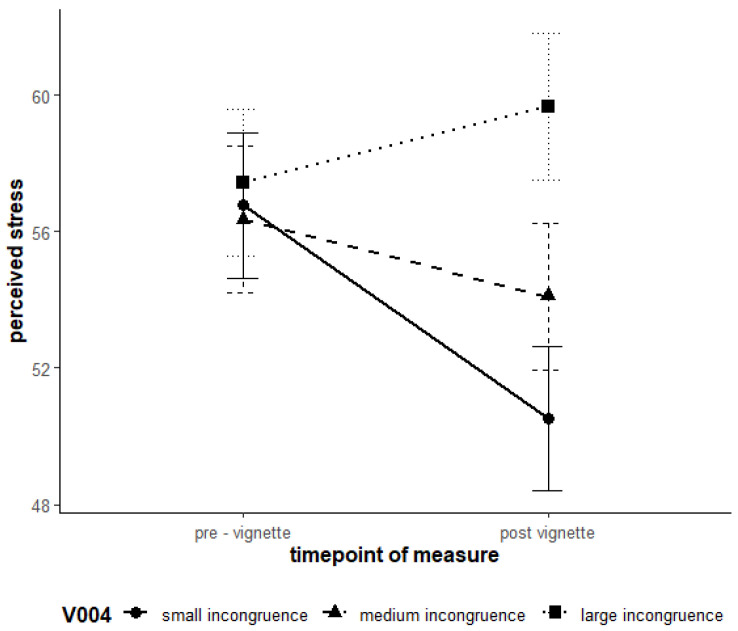
Interaction effects of time and the three incongruence groups on stress regarding the “Dependence” vignettes. Note. Error bars represent 95% confidence intervals.

**Figure 4 ejihpe-15-00128-f004:**
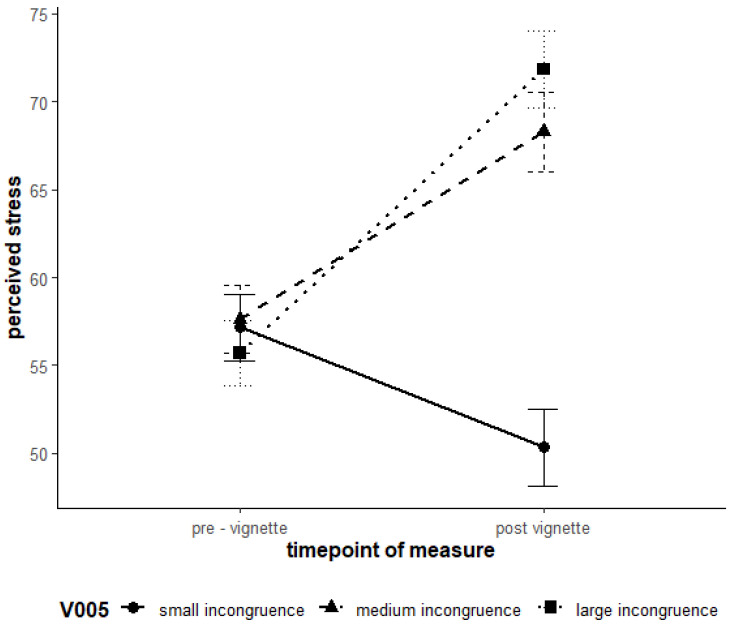
Interaction effects of time and the three incongruence groups on stress regarding the “Loss of Control” vignettes. Note. Error bars represent 95% confidence intervals.

**Table 1 ejihpe-15-00128-t001:** Overview of the pilot study results.

		Correct Assignment to Avoidance Goal			Correct Assignment to the Incongruence Situation		
Vignette		Count	Count in %	X^2^	Count	Count in %	X^2^
*N*		148			224		
Failure		111	72.9	277.75 ***			
	G1				181	80.8	232.03 ***
	G2				157	70.1	148.56 ***
	G3				184	82.1	244 ***
Devaluation		120.3	81.3	34.51 ***			
	G1				153	63.3	135.65 ***
	G2				162	72.3	153.25 ***
	G3				171	76.3	188.85 ***
Dependence		104.67	70.7	240.19 ***			
	G1				179	79.9	221.63 ***
	G2				156	69.6	133.56 ***
	G3				152	67.9	122.82 ***
Loss of Control		106.3	71.9	255.82 ***			
	G1				164	73.2	165.57 ***
	G2				151	67.4	121.94 ***
	G3				176	78.6	211.54 ***

Notes: *N* = number of participants; G1 = no to low incongruence, G2 = moderate incongruence, G3 = high incongruence; χ^2^ = chi-square test for probabilities; *** *p* < 0.001.

**Table 2 ejihpe-15-00128-t002:** Overview of model information and deviance measures for the dependent variable stress.

	Nullmodel					Model A: Interaction			Model B: Random Intercept			Model C: AR(1)-Structure	R^2^
	AIC	BIC	LogLik	Df	ICC	LogLik	L.Ratio	Df	LogLik	L.Ratio	Df	LogLik	
**Failure**					0.266								
Failure	7759.09	7759.09	−3876.55	3		−3660.23	**432.64 *****	14	−3660.11	0.22	16	−3660.23	0.34
Conflict Schema	7759.09	7773.63	−3876.55	3		−3666.99	**419.11 *****	14	−3660.11	13.76	16	−3666.99	0.33
**Devaluation**					0.211								
Devaluation	7884.824	7899.32	−3939.41	3		−3639.06	**600.71 *****	14	−3638.87	0.36	16	−3639.06	0.44
Conflict Schema	7884.824	7899.32	−3939.41	3		−3638.79	**601.24 *****	14	−3638.87	0.16	16	−3638.79	0.44
**Dependance**					0.447								
Dependance	7322.78	7337.25	−3658.39	3		−3629.04	**58.71 *****	14	−3629.03	0.01	16	−3629.04	0.05
Conflict Schema	7322.78	7337.25	−3658.39	3		−3628.61	**59.56 *****	14	−3629.03	0.84	16	−3628.61	0.05
**Loss of Control**					0.313								
Loss of Control	7574.95	7589.39	−3784.47	3		−3581.36	406.22 ***	14	−3575.25	**12.22 *****	16	−3581.36	0.27
Conflict Schema	7574.95	7589.39	−3784.47	3		−3582.61	403.72	14	−3575.25	**14.72 *****	16	−3582.61	0.34

Notes: The final model used is highlighted in bold. AIC = Akaike’s Information Criterion. BIC = Bayesian Information Criterion. L. Ratio = Log-likelihood values of two nested models were converted to deviance (× −2) and then differenced. The resulting *p*-value indicates whether a model fits the data significantly better than the model in the previous row. *** = *p* < 0.001. df = degrees of freedom. R^2^ according to [59] ([59]), indicates the proportion of variance explained by the fixed effects. based on [14] ([14]), variance explanation is considered small = 0.02, moderate = 0.13, large = 0.26. ICC = intraclass correlation coefficient.

**Table 3 ejihpe-15-00128-t003:** Overview of model information and deviance measures for the dependent variable wellbeing.

	Nullmodel					Model A: Interaction			Model B: Random Intercept			Modell C: AR(1)-Structure	R^2^
	AIC	BIC	LogLik	Df	ICC	LogLik	L.Ratio	Df	LogLik	L.Ratio	Df	LogLik	
**Failure**					0.553								
Failure	6292.181	6306.72	−3143.10	3		−3096.66	92.87 ***	14	−3082.94	**27.44 *****	16	−3096.65	0.10
Conflict Schema	6292.181	6306.72	−3143.10	3		−3082.94	92.87 ***	14	−3079.95	**33.41 *****	16	−3092.92	0.10
**Devaluation**					0.417								
Devaluation	6336.932	6351.43	−3165.47	3		−3107.52	115.90 ***	14	−3093.70	**27.65 *****	16	−3107.51	0.15
Conflict Schema	6336.932	6351.43	−3165.47	3		−3107.52	115.90 ***	14	−3092.12	**30.80 *****	16		0.10
**Dependance**					0.556								
Dependance	5994.96	6009.43	−2994.48	3		−2986.03	16.90	14	−2986.03	**5.75 ***	16	−2986.02	0.02
Conflict Schema	5994.96	6009.41	−2994.48	3		−2986.50	15.95	14	−2983.45	**6.11 ****	16	−2986.50	0.02
**Loss of Control**					0.381								
Loss of Control	6360.36	6374.81	−3177.18	3		−3113.15	128.07 ***	14	−3074.91	**76.47 *****	16	−3113.15	0.13
Conflict Schema	6360.36	6374.81	−3177.18	3		−3114.22	125.92 ***	14	−3078.22	**72.00 *****	16	−3114.22	0.13

Notes: The final model used is highlighted in bold. AIC = Akaike’s Information Criterion. BIC = Bayesian Information Criterion. L. Ratio = Log-likelihood values of two nested models were converted to deviance (× −2) and then differenced. The resulting *p*-value indicates whether a model fits the data significantly better than the model in the previous row. * = *p* < 0.10; ** = *p* < 0.05; *** = *p* < 0.001; df = degrees of freedom. R^2^ according to [59] ([59]), indicates the proportion of variance explained by the fixed effects. based on [14] ([14]), Variance explanation is considered small = 0.02, moderate = 0.13, large = 0.26. ICC = intraclass correlation coefficient.

**Table 4 ejihpe-15-00128-t004:** Overview of means and standard deviations of the collected interval-scaled measures for the dependent variables.

Variables		Time of Measurement 1		Time of Measurement 2	
		M	SD	M	SD
Dependant Variables					
SSSQ-G Failure					
	G1	56.18	13.65	54.72	12.11
	G2	56.29	12.62	68.49	13.19
	G3	58.29	13.41	75.99	13.78
SSSQ-G Devaluation					
	G1	56.04	12.69	45.84	11.18
	G2	56.19	12.32	71.55	12.66
	G3	58.28	14.72	79.66	14.81
SSSQ-G Dependence					
	G1	56.74	13.15	50.45	12.20
	G2	56.42	13.78	54.05	14.38
	G3	57.36	13.17	59.72	13.78
SSSQ-G Loss of Control					
	G1	56.99	13.90	50.29	13.69
	G2	57.64	13.55	68.21	14.52
	G3	55.78	12.53	71.83	15.58
PANAS_n Failure					
	G1	14.57	6.42	14.75	6.20
	G2	14.87	5.93	17.06	8.09
	G3	16.29	7.42	18.93	9.61
PANAS_n Devaluation					
	G1	14.92	6.45	12.51	4.48
	G2	15.17	5.98	16.92	8.15
	G3	15.47	7.19	20.07	10.62
PANAS_n Dependence					
	G1	14.42	6.03	14.17	6.54
	G2	15.59	7.15	15.71	7.42
	G3	15.41	6.36	16.39	7.64
PANAS_n Loss of Control					
	G1	14.58	5.77	14.33	7.12
	G2	15.44	7.04	19.31	10.18
	G3	15.40	6.77	19.52	10.01

Notes: For the calculation of means (M) and standard deviations (SD), missing values were excluded listwise. G1 = no to low discrepancy, G2 = moderate discrepancy, G3 = high discrepancy.

**Table 5 ejihpe-15-00128-t005:** Overview of means and standard deviations of the collected interval-scaled measures for the avoidance goals and conflict schema.

Variable		Time of Measurement 1		Time of Measurement 2	
		M	SD	M	SD
Performance/Failure					
Failure					
	G1	3.69	0.74	3.70	0.74
	G2	3.82	0.68	3.83	0.68
	G3	3.74	0.76	3.76	0.77
Conflict schema					
	G1	14.20	4.19	14.24	4.19
	G2	14.98	4.49	15.05	4.49
	G3	14.53	4.45	14.62	4.48
Recognition/Devaluation					
Devaluation					
	G1	3.84	0.72	3.84	0.72
	G2	3.97	0.65	3.98	0.65
	G3	3.97	0.75	3.97	0.75
Conflict schema					
	G1	15.81	4.53	15.81	4.54
	G2	16.62	4.53	16.71	4.52
	G3	16.87	4.74	16.85	4.74
Independence/Dependence					
Dependence					
	G1	4.01	0.70	52.91	15.29
	G2	3.93	0.68	65.85	13.65
	G3	4.06	0.73	68.11	15.95
Conflict schema					
	G1	17.35	4.79	17.34	4.79
	G2	16.92	4.36	16.89	4.37
	G3	17.81	4.65	17.89	4.58
Control/Loss of Control					
Loss of control					
	G1	2.59	0.89	2.59	0.89
	G2	2.62	0.91	2.62	0.91
	G3	2.62	0.93	2.62	0.93
Conflict schema					
	G1	10.33	4.26	10.36	4.28
	G2	10.73	4.45	10.74	4.43
	G3	10.61	4.57	10.62	4.57

Notes: For the calculation of means (M) and standard deviations (SD), missing values were excluded listwise. G1 = no to low discrepancy, G2 = moderate discrepancy, G3 = high discrepancy.

**Table 6 ejihpe-15-00128-t006:** Overview of multilevel models and estimation of fixed parameters for Model “Failure”.

	Stress						Wellbeing					
Model	Value	t ^a^	*p* ^b^	df	SE ^c^	f^2^	Value	t ^a^	*p* ^b^	df	SE ^c^	f^2^
Intercept	35.190	13.43	<0.001	471	0.99		14.727	28.766	<0.001	471	0.52	
Failure ^d^	6.160	8.70	<0.001	471	1.33	0.22	1.736	2.520	0.012	471	0.70	0.06
Conflict Schema ^d^	1.660	4.32	<0.001	471	0.24	0.19	0.341	2.795	0.02	471	0.14	0.07
Time of Measurement (1)	13.782	8.903	<0.001	457	1.57	0.10	1.995	2.565	0.006	457	0.75	0.006
Group (2)	19.252	8.903	<0.001	457	1.57	0.10	2.458	3.147	0.002	457	0.78	0.006
Group (3)	3.065	1.869	0.163	457	2.19		0.784	0.718	0.498	457	1.16	
Time of Measurement × Group (2) ^e^	1.380	0.663	0.507	457	2.08		−0.278	−0.268	0.777	457	0.98	
Time of Measurement × Group (3) ^e^	0.715	1.994	0.05	457	0.36		0.229	0.179	0.184	457	0.18	

Notes: *N* = 477 participants. 2 measurement points. 3 groups (low, moderate, high incongruence). ^a^ = t-statistic. ^b^ = significance. ^c^ = standard error of the mean. ^d^ = grand mean centered. ^e^ = Interaction effect between Time of Measurement and Group. f^2^ = effect size according to [56] ([56]), small = 0.02, medium = 0.15, large = 0.35 ([15]). reported only for relevant estimates.

**Table 7 ejihpe-15-00128-t007:** Overview of multilevel models and estimation of fixed parameters for Model “Devaluation”.

	Stress						Wellbeing					
Model	Value	t ^a^	*p* ^b^	df	SE ^c^	f^2^	Value	t ^a^	*p* ^b^	df	SE ^c^	f^2^
Intercept	56.765	52.620	<0.001	456	1.08		14.434	27.553	<0.001	456	0.48	
Devaluation ^d^	−0.133	−0.086	0.931	456	1.54	0.02	−0.637	−0.847	0.398	456	0.89	0.003
Conflict Schema ^d^	−0.221	−0.979	0.328	456	0.22	0.02	−0.186	−1.690	0.195	456	0.14	0.006
Time of Measurement (1)	3.989	2.513	<0.001	451	1.59	0.02	0.365	0.489	0.628	451	0.71	0.001
Group (2)	8.467	5.333	<0.001	451	1.59	0.02	1.186	1.585	0.091	451	0.69	0.001
Group (3)	3.272	1.419	0.157	451	2.31		0.584	0.539	0.590	451	1.08	
Time of Measurement × Group (2) ^e^	4.204	1.891	0.059	451	2.22		1.028	0.984	0.326	451	1.04	
Time of Measurement × Group (3) ^e^	0.313	0.8995	0.369	451	0.35		0.001	0.009	0.994	451	0.18	

Notes: *N* = 468 participants. 2 measurement points. 3 groups (low, moderate, high incongruence). ^a^ = t-statistic. ^b^ = significance. ^c^ = standard error of the mean. ^d^ = grand mean centered. ^e^ = Interaction effect between Time of Measurement and Group. f^2^ = effect size according to [56] ([56]). small = 0.02, medium = 0.15, large = 0.35 ([15]). reported only for relevant estimates.

**Table 8 ejihpe-15-00128-t008:** Overview of multilevel models and estimation of fixed parameters for Model “Dependence”.

	Stress						Wellbeing					
Model	Value	t ^a^	*p* ^b^	df	SE ^c^	f^2^	Value	t ^a^	*p* ^b^	df	SE ^c^	f^2^
Intercept	56.765	52.620	<0.001	456	1.08		14.434	27.553	<0.001	456	0.48	
Dependence ^d^	−0.133	−0.086	0.931	456	1.54	0.02	−0.637	−0.847	0.398	456	0.89	0.003
Conflict Schema ^d^	−0.221	−0.979	0.328	456	0.22	0.02	−0.186	−1.690	0.195	456	0.14	0.006
Time of Measurement (1)	3.989	2.513	<0.001	451	1.59	0.02	0.365	0.489	0.628	451	0.71	0.001
Group (2)	8.467	5.333	<0.001	451	1.59	0.02	1.186	1.585	0.091	451	0.69	0.001
Group (3)	3.272	1.419	0.157	451	2.31		0.584	0.539	0.590	451	1.08	
Time of Measurement × Group (2) ^e^	4.204	1.891	0.059	451	2.22		1.028	0.984	0.326	451	1.04	
Time of Measurement × Group (3) ^e^	0.313	0.8995	0.369	451	0.35		0.001	0.009	0.994	451	0.18	

Notes: *N* = 462 participants. 2 measurement points. 3 groups (low, moderate, high incongruence). ^a^ = t-statistic. ^b^ = significance. ^c^ = standard error of the mean. ^d^ = grand mean centered. ^e^ = Interaction effect between Time of Measurement and Group. f^2^ = effect size according to [56] ([56]). small = 0.02, medium = 0.15, large = 0.35 ([15]). reported only for relevant estimates.

**Table 9 ejihpe-15-00128-t009:** Overview of multilevel models and estimation of fixed parameters for Model “Loss of Control”.

	Stress						Wellbeing					
Model	Value	t ^a^	*p* ^b^	df	SE ^c^	f^2^	Value	t ^a^	*p* ^b^	df	SE ^c^	f^2^
Intercept	57.160	54.723	<0.001	456	0.98	0.30	14.726	29.319	<0.001	456	0.43	
Loss of Control ^d^	7.768	7.134	<0.001	456	1.14	0.30	2.779	4.949	<0.001	456	0.56	0.08
Conflict Schema ^d^	1.596	6.967	<0.001	456	0.25		0.559	4.733	<0.001	456	0.12	0.08
Time of Measurement (1)	17.513	10.642	<0.001	443	1.65	0.14	4.174	4.177	<0.001	456	0.92	0.02
Group (2)	22.974	14.203	<0.001	443	1.63	0.14	4.515	4.602	<0.001	456	0.95	0.02
Group (3)	1.643	0.898	0.399	443	1.94		4.167	4.180	0.087	443	1.21	
Time of Measurement × Group (2) ^e^	−0.404	−0.227	0.820	443	1.78		4.487	4.583	0.817	443	1.14	
Time of Measurement × Group (3) ^e^	0.462	1.216	0.225	443	0.43		0.441	1.919	0.109	443	0.27	

Notes: *N* = 462 participants. 2 measurement points. 3 groups (low, moderate, high incongruence). ^a^ = t-statistic. ^b^ = significance. ^c^ = standard error of the mean. ^d^ = grand mean centered. ^e^ = Interaction effect between Time of Measurement and Group. f^2^ = effect size according to [56] ([56]). small = 0.02, medium = 0.15, large = 0.35 ([15]). reported only for relevant estimates.

## Data Availability

Data supporting the conclusions of this article is available can be shared by the authors upon request.

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
