# Peer review of "The Hidden Cost of High Aspirations: Examining the Stress-Enhancing Effect of Motivational Goals Using Vignette Methodology"

_ejihpe, 2025, doi:10.3390/ejihpe15070128_

Round 1
Reviewer 1 Report
Comments and Suggestions for Authors
This article already has a suitability with the journal's special topic. Minor input is whether it is also possible for researchers to convey about the development of instruments that can be carried out by subsequent researchers who may be interested in developing research.
1. Main Question Addressed by the Research The study has successfully investigated whether motivational goals and internalized conflict schemas (based on Grawe’s Consistency Theory) intensify subjective stress when approach and avoidance goals are simultaneously activated. Specifically, this research examines:
- How avoidance goals and conflict schemas influence baseline stress and well-being.
- Whether experimentally induced incongruence (misalignment between goals and reality) increases stress responses in workplace scenarios.
2. Originality and Relevance The topic is original in its experimental approach to testing Grawe’s theory, which has primarily been studied correlationally (e.g., Fries & Grawe, 2006). The vignette methodology to manipulate incongruence is innovative. This study addresses a gap in stress research by focusing on internal psychological dynamics (e.g., motivational conflicts) rather than external stressors alone. This is critical for personalized stress interventions.
3. Contribution to the Field Compared to prior work, this study provides causal evidence linking motivational incongruence to stress, beyond correlational findings. Integrates Grawe’s Consistency Theory and Cybernetic Theory of Stress into occupational psychology, offering a unified framework. This article also highlights the role of social support as a buffer, even in high-incongruence scenarios (e.g., "Dependence" vignette).
4. Methodological Improvements The sample was 83% female and predominantly young students. Future studies should include broader demographics (age, gender, occupations) for generalizability. In this article stress was self-reported; adding cortisol levels or heart rate variability could strengthen validity. In this research, also highlights personality traits (e.g., neuroticism) or coping styles were omitted, which might confound results. We also may found the calming effect in control groups (e.g., stress reduction) suggests unintended regulatory effects. Pilot testing could refine vignettes to ensure neutrality.
5. Consistency of Conclusions Hypotheses H1–H3 were largely supported, showing avoidance goals and incongruence increase stress (except for "Dependence," where social support may have mitigated effects). The unexpected stress reduction in control groups is thoughtfully discussed as a potential regulatory effect of low-incongruence scenarios, linking to therapeutic literature (e.g., Berking et al., 2003).
6. Appropriateness of References Comprehensive coverage of stress theories (Grawe, Edwards), validated tools (FAMOS, SSSQ-G), and recent meta-analyses (e.g., Bhui et al., 2012). Some older citations (e.g., Grawe, 1998) dominate but this is acceptable; newer work on motivational conflicts (e.g., 2010s+) could be included.
7. Comments on Tables and Figures In tables 2–3, clear means/SD for stress/well-being across conditions, but could be condensed. In figures 1–4 mentioned its effectively show stress changes by incongruence level. Add error bars to clarify variability. Label y-axes consistently (e.g., "Stress Level" vs. "Perceived Stress"). Appendix A1: Vignette texts are well-detailed and validated (pilot study).
8. Additional Comments The discussion excellently ties findings to clinical (e.g., psychotherapy) and organizational (e.g., job resources) contexts. In practical recommendations, the guided imagery (GI) is a novel suggestion for stress management, but feasibility in workplaces needs elaboration. Acknowledge potential bias from student sample and self-report data.
Author Response
Reviewer 1 / Comment 1:
This article already has a suitability with the journal's special topic. Minor input is whether it is also possible for researchers to convey about the development of instruments that can be carried out by subsequent researchers who may be interested in developing research.
Response to Reviewer 1 / Comment 1:
In light of the limitations discussed, future research could focus on the development of standardized instruments based on the vignette methodology used in this study. The current vignettes employed brief narrative scenarios designed to activate motivational incongruence and induce stress. These narrative prompts could be systematically refined in terms of content, length, and emotional intensity to ensure reliable elicitation of targeted motivational drivers (e.g., fear of failure, excessive control needs, dependency). For these purposes, these vignettes are included in the appendix and can be used and further developed by researchers.
Reviewer 1 / Comment 2:
The sample was 83% female and predominantly young students. Future studies should include broader demographics (age, gender, occupations) for generalizability.
Response to Reviewer 1 / Comment 2:
Thank you for your valuable comment. We acknowledge that the sample was predominantly female (83%) and young, which may limit the generalizability of the results to other demographic groups. This limitation has been discussed in the study on page 17: “The sample consisted primarily of young female university students (mean age 25 years), which may limit the generalizability of the findings. However, the participants ranged in age from 16 to 63 years, and 71% were employed either part-time or full-time. These factors suggest that, despite the predominance of students, the sample still reflects a relatively diverse and working population. Future studies should include more diverse samples across gender, age, and occupational backgrounds”.
Reviewer 1 / Comment 3:
In this article stress was self-reported; adding cortisol levels or heart rate variability could strengthen validity.
Response to Reviewer 1 / Comment 3:
Thank you for your valuable comment. We recognize that self-report data on stress are susceptible to biases such as social desirability or self-perception errors. In this study, self-report measures were used due to their practicality and common use in similar research. However, we agree that incorporating physiological measures such as cortisol levels or heart rate variability would enhance the validity of stress assessments. This limitation has been mentioned in the study on page 17: “Additionally, stress was assessed via self-report only, without physiological measures such as cortisol, which could improve validity”. For future studies, we intend to integrate these objective measures to provide a more comprehensive understanding of stress responses.
Reviewer 1 / Comment 4:
This research also highlights that personality traits (e.g., neuroticism) or coping styles were omitted, which might confound results.
Response to Reviewer 1 / Comment 4:
Thank you for your valuable comment. We agree that the omission of personality traits (e.g., neuroticism) and coping styles could have influenced the results, as noted in the study on page 17: “Control variables such as personality traits or coping styles were not included, which may have influenced outcomes.” These factors may indeed play a moderating role in stress responses, and their exclusion was a limitation of the current study. In future research, we plan to incorporate these variables to better control for potential confounding effects and to gain a deeper understanding of how individual differences influence the outcomes.
Reviewer 1 / Comment 5:
We also may found that the calming effect in control groups (e.g., stress reduction) suggests unintended regulatory effects. Pilot testing could refine vignettes to ensure neutrality.
Response to Reviewer 1 / Comment 5:
Thank you for your valuable comment. We acknowledge that the calming effect observed in the control group could suggest unintended regulatory effects due to the vignette format itself as mentioned on page 17 “Although manipulation checks revealed no systematic bias, the calming effect in the control condition may point to an unintentional regulatory function of the vignette format itself.”. While manipulation checks indicated no systematic bias, we recognize that the vignette format may have inadvertently led to calming effects in the control group. To address this, we suggest conducting pilot testing in future studies to refine the vignettes and ensure their neutrality, which will help eliminate any unintended effects
Reviewer 1 / Comment 6:
Some older citations (e.g., Grawe, 1998) dominate but this is acceptable; newer work on motivational conflicts (e.g., 2010s+) could be included.
Response to Reviewer 1 / Comment 6:
Thank you for your valuable comment. Regarding Grawes Consistency Theory the newest research found was Fries et al. (2008). That's why we decided to do some more general research.
We found that recent research on goal conflict—such as studies by Sakaki et al, (2024)—offers valuable insights into how avoidance goals can undermine well-being and motivation. However, while the concept of approach and avoidance goals is present in a lot of found studies, it stems from different theoretical traditions (for example BAS/BIS framework, Achievement Goal Theory, job crafting) and is defined inconsistently across studies (Sakaki et al, 2024; Ebert et al, 2024; ÅšwiÄ…tkowski & Dompnier, 2021. These approaches differ from Grawe’s Consistency Theory in several ways. Most notably, they often emphasize context-specific, externally induced conflicts (e.g., workload, role demands), whereas Grawe focuses on internal motivational structures and perceived discrepancies.
These conceptual inconsistencies highlight that while modern research aligns with some principles of Consistency Theory—especially regarding stress resulting from blocked or conflicting goals—it often lacks a unified motivational framework and does not fully account for the deeper psychological dynamics these classic theories propose.
Newer findings of Gorges et al. (2022) highlight the critical role of internal motivational dynamics—particularly goal conflicts—in contributing to psychological strain. Their research demonstrated that conflicting goals in the teaching profession undermine basic psychological needs, especially autonomy and competence, leading to reduced autonomous motivation and increased stress. This supports the broader theoretical framework suggesting that motivational inconsistency—whether structural or experimentally induced—plays a key role in the development of work-related stress.
Only a few studies have applied the cybernetic theory of stress, spanning from 2007 to 2018 (e.g., Franco & Trombetta, 2010; Stich et al., 2018). While their findings align with the general conclusions of this study, they primarily focus on external stressors rather than motivational goals. In organizational psychology research, perceived discrepancies between desired and actual work conditions—so-called "GAPs"—have been shown to predict well-being more accurately than perceptions alone (Franco & Trombetta, 2010). Similarly, Stich et al. (2018) used cybernetic theory to demonstrate that individuals manage their email use through discrepancy-reducing and discrepancy-enhancing mechanisms, actively adjusting their desired email load in response to workload stress. This underscores the role of self-regulation in coping with email overload, beyond simply experiencing stress.
We therefore added a paragraph into the introduction which explains the state of research in a somewhat broader context in the introduction in order to highlight the research gap (introduction, p. 2-3):
“Instead, there are a few indirect indications from various studies that broadly support the assumptions. Research on approach and avoidance goals by Sakaki et al. (2024) of-fers insights into how avoidance goals can undermine well-being and motivation. However, most studies (e.g., Sakaki et al., 2024; Ebert et al., 2024; ÅšwiÄ…tkowski & Dompnier, 2021) focus on externally induced, context-specific conflicts (e.g., workload, role demands), diverging from the emphasis on internal goal congruence. Although these studies incorporate the concept of approach and avoidance goals, they do so from diverse theoretical traditions (such as the BAS/BIS framework, Achievement Goal Theory, or job crafting), and definitions vary considerably across studies. This concep-tual inconsistency limits the ability to unify findings under a common motivational framework. Some studies, such as Gorges et al. (2022), do address internal motivation-al dynamics more directly. Their findings show that goal conflicts among teachers un-dermine basic psychological needs like autonomy and competence, reducing autono-mous motivation and increasing stress. This supports the broader idea that motiva-tional inconsistencies contribute to psychological strain. A small number of studies drawing on cybernetic theory (e.g., Franco & Trombetta, 2010; Stich et al., 2018) offer related insights. They show that perceived discrepancies between actual and desired work conditions predict well-being more accurately than subjective perceptions alone and highlight self-regulatory mechanisms in coping with stress (e.g., in email over-load).
In summary, while contemporary research provides valuable findings that are compatible with certain aspects of Consistency Theory, there remains a lack of empir-ical studies that directly test its central theoretical assumption: that the perceived con-sistency of internal motivational structures is a key determinant of psychological well-being.
Although the relationship between motivational goals and stress has been ad-dressed in previous correlational research (e.g., Fries & Grawe, 2006), a systematic ex-perimental investigation is still lacking. Therefore, the aim of the present study is to address this gap by experimentally testing whether frustrated motivational goals—operationalized through psychologically relevant vignettes—lead to increases in subjective stress. By applying a randomized vignette design with pre- and post-stress assessments, this study provides an important step toward clarifying the internal mechanisms of stress responses, and informing more individualized strategies for occupational health and well-being.”
References:
Ebert, T., Först, R., & Bipp, T. (2024). The dark and potentially bright sides of work-avoidance goal orientation. Frontiers in Organizational Psychology, 2, 1445014. https://doi.org/10.3389/forgp.2024.1445014
Franco, M., & Trombetta, M. (2010). Cybernetic theory as a new approach to studying workers’ well-being. Croatian Economic Survey, 12(1), 27–55. URI: https://hrcak.srce.hr/52494
Gorges, J., Neumann, P., & Störtländer, J. C. (2022). Teachers between a rock and a hard place: Goal conflicts affect teaching motivation mediated by basic need satisfaction. Frontiers in Psychology, 13, 876521. https://doi.org/10.3389/fpsyg.2022.876521
Sakaki, M., Murayama, K., Izuma, K., Aoki, R., Yomogita, Y., Sugiura, A., Singhi, N., Matsumoto, M., & Matsumoto, K. (2024). Motivated with joy or anxiety: Does approach-avoidance goal framing elicit differential reward-network activation in the brain? Cognitive, Affective & Behavioral Neuroscience, 24(3), 469–490. https://doi.org/10.3758/s13415-024-01154-3
Stich, J. F., Tarafdar, M., Stacey, P., & Cooper, C. L. (2019). E-mail load, workload stress and desired e-mail load: a cybernetic approach. Information Technology & People, 32(2), 430-452. https://doi.org/10.1108/ITP-10-2017-0321
ÅšwiÄ…tkowski, W., & Dompnier, B. (2021). When pursuing bad goals for good reasons makes it even worse: A social value approach to performance-avoidance goal pursuit. Social Psychology of Education, 24(3), 653-677. https://doi.org/10.1007/s11218-021-09623-0
Reviewer 1 / Comment 7:
In tables 2–3, clear means/SD for stress/well-being across conditions, but could be condensed. In figures 1–4 mentioned it effectively shows stress changes by incongruence level. Add error bars to clarify variability. Label y-axes consistently (e.g., "Stress Level" vs. "Perceived Stress").
Response to Reviewer 1 / Comment 7:
Thank you for your valuable comment. Variability in the data is already illustrated by the error bars displayed in the figure. To address your comment and improve clarity for the reader, a note was added to the figure caption stating that the error bars represent 95% confidence intervals. We furthermore revised the figures, the y-axes are now labeled consistently “perceived stress”.
Reviewer 1 / Comment 8:
The discussion excellently ties findings to clinical (e.g., psychotherapy) and organizational (e.g., job resources) contexts. In practical recommendations, the guided imagery (GI) is a novel suggestion for stress management, but feasibility in workplaces needs elaboration. Acknowledge potential bias from student sample and self-report data.
Response to Reviewer 1 / Comment 8:
Thank you for your valuable comment. We rewrote the part of the discussion regarding GI as follows (p. 19) to explain the topic in more detail:
“Third, Guided Imagery (GI) could be leveraged to address motivational schemas, particularly in managing identified avoidance goals or conflict schemas, within the framework of stress management interventions or leadership development programs. In Stress Management Programs, GI can help employees address avoidance goals and conflict schemas by visualizing calm, controlled environments and successfully navigating challenging situations. To further enhance the effectiveness of GI in the workplace, companies could consider integrating it through mobile apps or online platforms, allowing employees to engage in guided imagery exercises tailored to specific scenarios such as "relaxation before a presentation" or "calming after a stressful meeting." Additionally, companies could create designated relaxation rooms equipped with audio or video-guided imagery sessions, offering employees a dedicated space for short breaks during the day. Incorporating these practices during lunch breaks could also provide a valuable opportunity for employees to recharge and relieve stress. However, while GI is a novel suggestion for stress management, its feasibility in workplace settings needs further elaboration. Practical considerations, such as the work environment, employee engagement, and potential barriers to implementation, should be explored to determine how effectively GI can be incorporated into workplace stress interventions.”
Furthermore, we acknowledge that the sample predominantly consisted of young female university students (mean age 25 years), which may limit the generalizability of the findings, particularly since students may have different experiences with occupational stress compared to full-time employees. However, the participants ranged from 16 to 63 years of age, and 71% of the participants were employed either part-time or full-time. While we recognize that the student population is overrepresented, we believe that the inclusion of employed individuals adds diversity to the sample. For future studies, we recommend recruiting a broader sample of full-time employees, career professionals, and a more balanced representation of students to improve the generalizability of the findings.
Furthermore, regarding limitations and study participants we added some information. We added these information to the manuscript in the methods section (p. 7; 3.3. Study Participants) and in the limitations (p. 19f):
p. 7: “Participants were recruited via a link from students in distance learning programs at the International University of Applied Sciences (IU) and via social media channels. Students in distance learning programs are often part-time students, balancing their studies alongside full-time jobs.. Therefore, these students typically have significant work experience. Furthermore, work experience was an inclusion criterium for the study: To participate, respondents had to be at least 18 years old, not undergoing psychological treatment, and have work experience.”
p. 19: “The sample consisted predominantly of young female university students (mean age 25 years), which may limit the generalizability of the findings, particularly since students may have different experiences with occupational stress compared to full-time employees. This may limit the generalizability of the findings. However, the participants’ age ranged from 16 to 63 years, and 71% of the participants were employed either part-time or full-time. While the student population is overrepresented, the inclusion of employed individuals adds diversity to the sample. For future studies, it is recommended to recruite a broader sample of employees, and a more balanced representation of students to improve the generalizability of the findings.
[…] In addition, the reliance on self-report data for assessing stress may introduce common method bias. Although our pre–post experimental design and randomization strengthen causal inference and mitigate endogeneity concerns, self-reports remain susceptible to shared method variance. Moreover, the simultaneous measurement of variables within one session may lead to inflated associations. To address this, future studies should consider using multiple time points—e.g., measuring certain constructs well before or after the intervention—to disentangle temporal effects more clearly. Furthermore, incorporating physiological indicators (e.g., cortisol) or observer-rated assessments would provide more robust evidence and increase validity.”
Reviewer 2 Report
Comments and Suggestions for Authors
This is an interesting study. This study investigates whether motivational goals and internalized conflict schemas—as proposed by Grawe’s Consistency Theory—account for these differences by
intensifying subjective stress when approach and avoidance goals are simultaneously activated. Relevant literature has been used. Better clarify the literature gap. It would be better not to use full time student samples since the topic is related to occupational stress. Students may not care about the part-time job. Might need to check using multi-group analysis. How to deal with common method bias and endogeneity? Why collect views from a Germany university? Is it represent the other universities all over the world? Authors need to compare the results of related studies in discussion.
Author Response
Reviewer 2 / Comment 1:
This is an interesting study. This study investigates whether motivational goals and internalized conflict schemas—as proposed by Grawe’s Consistency Theory—account for these differences by intensifying subjective stress when approach and avoidance goals are simultaneously activated. Relevant literature has been used. Better clarify the literature gap.
Response to Reviewer 2 / Comment 1:
Thank you for your constructive feedback and for highlighting the importance of clarifying the literature gap. We have revised the relevant section to better articulate what previous research has already shown and where there are still gaps. We added some more studies on the topic in general which better explain the research gap. For example we added recent research on goal conflict—such as studies by Sakaki et al, (2024)— which offers valuable insights into how avoidance goals can undermine well-being and motivation. However, while the concept of approach and avoidance goals is present in a lot of found studies, it stems from different theoretical traditions (for example BAS/BIS framework, Achievement Goal Theory, job crafting) and is defined inconsistently across studies (Sakaki et al, 2024; Ebert et al, 2024; ÅšwiÄ…tkowski & Dompnier, 2021. These approaches differ from Grawe’s Consistency Theory in several ways. Most notably, they often emphasize context-specific, externally induced conflicts (e.g., workload, role demands), whereas Grawe focuses on internal motivational structures and perceived discrepancies. These conceptual inconsistencies highlight that while modern research aligns with some principles of Consistency Theory—especially regarding stress resulting from blocked or conflicting goals—it often lacks a unified motivational framework and does not fully account for the deeper psychological dynamics these classic theories propose.
Furthermore, newer findings of Gorges et al. (2022) highlight the critical role of internal motivational dynamics—particularly goal conflicts—in contributing to psychological strain. Their research demonstrated that conflicting goals in the teaching profession undermine basic psychological needs, especially autonomy and competence, leading to reduced autonomous motivation and increased stress. This supports the broader theoretical framework suggesting that motivational inconsistency—whether structural or experimentally induced—plays a key role in the development of work-related stress.
Only a few studies have applied the cybernetic theory of stress, spanning from 2007 to 2018 (e.g., Franco & Trombetta, 2010; Stich et al., 2018). While their findings align with the general conclusions of this study, they primarily focus on external stressors rather than motivational goals. In organizational psychology research, perceived discrepancies between desired and actual work conditions—so-called "GAPs"—have been shown to predict well-being more accurately than perceptions alone (Franco & Trombetta, 2010). Similarly, Stich et al. (2018) used cybernetic theory to demonstrate that individuals manage their email use through discrepancy-reducing and discrepancy-enhancing mechanisms, actively adjusting their desired email load in response to workload stress. This underscores the role of self-regulation in coping with email overload, beyond simply experiencing stress.
Concluding, we therefore added a paragraph into the introduction which explains the state of research in a somewhat broader context in the introduction in order to better highlight the research gap (see introduction, p. 2-3):
“Instead, there are a few indirect indications from various studies that broadly support the assumptions. Research on approach and avoidance goals by Sakaki et al. (2024) of-fers insights into how avoidance goals can undermine well-being and motivation. However, most studies (e.g., Sakaki et al., 2024; Ebert et al., 2024; ÅšwiÄ…tkowski & Dompnier, 2021) focus on externally induced, context-specific conflicts (e.g., workload, role demands), diverging from the emphasis on internal goal congruence. Although these studies incorporate the concept of approach and avoidance goals, they do so from diverse theoretical traditions (such as the BAS/BIS framework, Achievement Goal Theory, or job crafting), and definitions vary considerably across studies. This concep-tual inconsistency limits the ability to unify findings under a common motivational framework. Some studies, such as Gorges et al. (2022), do address internal motivation-al dynamics more directly. Their findings show that goal conflicts among teachers un-dermine basic psychological needs like autonomy and competence, reducing autono-mous motivation and increasing stress. This supports the broader idea that motiva-tional inconsistencies contribute to psychological strain. A small number of studies drawing on cybernetic theory (e.g., Franco & Trombetta, 2010; Stich et al., 2018) offer related insights. They show that perceived discrepancies between actual and desired work conditions predict well-being more accurately than subjective perceptions alone and highlight self-regulatory mechanisms in coping with stress (e.g., in email over-load).
In summary, while contemporary research provides valuable findings that are compatible with certain aspects of Consistency Theory, there remains a lack of empir-ical studies that directly test its central theoretical assumption: that the perceived con-sistency of internal motivational structures is a key determinant of psychological well-being.
Although the relationship between motivational goals and stress has been ad-dressed in previous correlational research (e.g., Fries & Grawe, 2006), a systematic ex-perimental investigation is still lacking. Therefore, the aim of the present study is to address this gap by experimentally testing whether frustrated motivational goals—operationalized through psychologically relevant vignettes—lead to increases in subjective stress. By applying a randomized vignette design with pre- and post-stress assessments, this study provides an important step toward clarifying the internal mechanisms of stress responses, and informing more individualized strategies for occupational health and well-being.”
Reviewer 2 / Comment 2:
It would be better not to use full-time student samples since the topic is related to occupational stress. Students may not care about the part-time job.
Response to Reviewer 2 / Comment 2:
Thank you for your valuable comment. We understand the concern that full-time students might not have the same perspective on occupational stress as individuals in full-time employment. It is important to note that many students at the IU (distance university) are part-time students, often balancing their studies alongside full-time jobs. This makes the sample from a distance university particularly relevant, as these students typically have significant work experience. Furthermore, participants were specifically instructed that they should have work experience in order to be eligible for the study. While we acknowledge that the experiences of full-time students may differ from those of full-time employees, we believe the sample provides a valuable reflection of individuals who are actively engaged in both academic and professional settings. To address this limitation in future research, we suggest focusing on a broader range of full-time employees and early-career professionals to better align with the topic of occupational stress.
We added these information to the manuscript in the methods section (p. 7; 3.3. Study Participants) and in the limitations (p. 19):
p. 7: “Participants were recruited via a link from students in distance learning programs at the International University of Applied Sciences (IU) and via social media channels. Students in distance learning programs are often part-time students, balancing their studies alongside full-time jobs.. Therefore, these students typically have significant work experience. Furthermore, work experience was an inclusion criterium for the study: To participate, respondents had to be at least 18 years old, not undergoing psychological treatment, and have work experience.”
p. 19: “The sample consisted predominantly of young female university students (mean age 25 years), which may limit the generalizability of the findings, particularly since students may have different experiences with occupational stress compared to full-time employees. This may limit the generalizability of the findings. However, the participants’ age ranged from 16 to 63 years, and 71% of the participants were employed either part-time or full-time. While the student population is overrepresented, the inclusion of employed individuals adds diversity to the sample. For future studies, it is recommended to recruite a broader sample of employees, and a more balanced representation of students to improve the generalizability of the findings.”
Reviewer 2 / Comment 3:
Might need to check using multi-group analysis.*
Response to Reviewer 2 / Comment 3:
Thank you for your valuable comment and the suggestion. We appreciate the reviewer’s suggestion to conduct a multi-group analysis. Due to sample size limitations in certain subgroups and the fact that we did not include it as a variable in this form, such an analysis was not feasible in the current study. Nevertheless, we consider this a valuable direction for future research to explore possible differences in stress perception and regulation between full-time employees and students working part-time.
Reviewer 2 / Comment 4:
How to deal with common method bias and endogeneity?
Response to Reviewer 2 / Comment 4:
We thank the reviewer for raising the issue of potential endogeneity and common method bias. Our study employed an experimental vignette design with randomized assignment and pre- and post-intervention stress assessments, which supports causal inference regarding the effect of the manipulation. Due to the exogenous treatment assignment, we consider endogeneity a negligible concern in our design.
Nonetheless, we acknowledge the potential for common method bias, as our data rely on self-reported measures collected within the same survey session. To validate the effect of the manipulation and address this concern, we conducted a multilevel analysis. The results revealed a significant increase in stress levels in the experimental vignette group from pre- to post-assessment, while a decrease was observed in the control group. This finding supports the conclusion that the manipulation had a genuine effect on participants’ stress responses, rather than reflecting method-related artifacts or general response tendencies.
To further reduce common method bias and strengthen the temporal robustness of findings, future studies could integrate additional measurement points, for example assessing certain variables either prior to or well after the intervention. Moreover, complementing self-reports with objective data sources, such as physiological indicators (e.g., cortisol), would improve construct validity. Nevertheless, we emphasize that the randomized assignment remains a key strength of our design, enabling a rigorous test of causal effects.
We added to the manuscript in the limitation section (p. 20-21):
“In addition, the reliance on self-report data for assessing stress may introduce common method bias. Although our pre–post experimental design and randomization strengthen causal inference and mitigate endogeneity concerns, self-reports remain susceptible to shared method variance. Moreover, the simultaneous measurement of variables within one session may lead to inflated associations. To address this, future studies should consider using multiple time points—e.g., measuring certain constructs well before or after the intervention—to disentangle temporal effects more clearly. Furthermore, incorporating physiological indicators (e.g., cortisol) or observer-rated assessments would provide more robust evidence and increase validity.”
Reviewer 2 / Comment 5:
Why collect views from a German university? Is it representative of other universities all over the world?
Response to Reviewer 2 / Comment 5:
Thank you for raising this important point. We agree that focusing on a sample from a single German university may limit the representativeness of the findings. While the sample is from one university, it is important to note that many students at universities with mainly distance learning programs balance their studies with full-time or part-time employment, providing a unique perspective on occupational stress. This aspect makes the sample particularly relevant, even though it is not globally representative. We acknowledge that the current findings should be considered preliminary and context-specific. However, the use of a randomized experimental design provides strong internal validity and supports initial causal conclusions. Future studies should replicate the findings in more diverse samples across regions, occupations, and cultural contexts to assess the robustness and generalizability of the effects.
We furthermore recognize that cultural and social differences may influence perceptions of stress (e. g., Anderson, 1989; Shavitt et al, 2016; Day & Livingstone, 2003; Meyer et al, 2008), and we suggest that future research include more diverse samples from universities in different countries to enhance the external validity and generalizability of the results. Expanding the sample geographically and culturally would provide a more comprehensive understanding of occupational stress across various contexts.
This aspect was also added to the manuscript in the limitations section (p. 20):
“Furthermore, cultural and social differences (Anderson, 1989; Day & Livingstone, 2003; Meyer et al, 2008; Shavitt et al, 2016) may influence perceptions of stress, and therefore, future research should also include more diverse samples from universities in different countries to enhance the external validity and generalizability of the results.”
References:
Anderson N. B. (1989). Racial differences in stress-induced cardiovascular reactivity and hypertension: Current status and substantive issues. Psychological Bulletin, 105(1), 89-105. https://doi.org/10.1037/0033-2909.105.1.89
Day, A. L., & Livingstone, H. A. (2003). Gender differences in perceptions of stressors and utilization of social support among university students. Canadian Journal of Behavioural Science / Revue canadienne des sciences du comportement, 35(2), 73–83. https://doi.org/10.1037/h0087190
Meyer, I. H., Schwartz, S., & Frost, D. M. (2008). Social patterning of stress and coping: Does disadvantaged social statuses confer more stress and fewer coping resources? Social Science & Medicine, 67(3), 368–379. https://doi.org/10.1016/j.socscimed.2008.03.012
Shavitt, S., Cho, Y. I., Johnson, T. P., Jiang, D., Holbrook, A., & Stavrakantonaki, M. (2016). Culture Moderates the Relation Between Perceived Stress, Social Support, and Mental and Physical Health. Journal of Cross-Cultural Psychology, 47(7), 956-980. https://doi.org/10.1177/0022022116656132
Reviewer 2 / Comment 6:
Authors need to compare the results of related studies in discussion.
Response to Reviewer 2 / Comment 6:
Thank you very much for your valuable comment. The present findings are consistent with previous research demonstrating the detrimental effects of motivational inconsistency on psychological functioning. A meta-analysis by Fries and Grawe (2006) confirmed significant associations between various forms of motivational inconsistency—such as goal incongruence, internal conflict (discordance), and avoidance goals—and increased psychological symptoms, including depression, anxiety, and somatization. Notably, these associations were identified across numerous correlational studies, underscoring the theoretical assumptions of Grawe’s Consistency Theory but leaving causal mechanisms untested.
The current study extends this body of research by providing experimental evidence that motivational incongruence, even when induced through hypothetical scenarios, leads to acute increases in subjective stress. This supports the claim that motivational inconsistency is not merely correlated with but can actively cause psychological strain.
Further support comes from a randomized controlled trial by Fries et al. (2008), which examined whether motivational inconsistency could serve as a differential indication for specific therapeutic approaches. Their results showed that individuals with high levels of avoidance goals and goal conflict benefited more from clarification-oriented psychotherapy compared to disorder-specific interventions. While their study focused on real-life, persistent motivational conflict, the present study demonstrates that even transient, situationally induced incongruence can elicit measurable stress responses.
Newer findings of Gorges et al. (2022) align well with the current results, as both highlight the critical role of internal motivational dynamics—particularly goal conflicts—in contributing to psychological strain. Their research demonstrated that conflicting goals in the teaching profession undermine basic psychological needs, especially autonomy and competence, leading to reduced autonomous motivation and increased stress. This supports the broader theoretical framework suggesting that motivational inconsistency—whether structural or experimentally induced—plays a key role in the development of work-related stress.
These findings are consistent with research from organizational psychology, where perceived discrepancies between desired and actual work conditions (so-called GAPs) have been shown to predict well-being more accurately than perceptions alone (Franco & Trombetta, 2010). This convergence of evidence from both motivational theory and workplace research underscores the pivotal role of discrepancy-based stress models, as described in the Cybernetic Theory of Stress (Edwards, 1992).
In the discussion section we therefore added this paragraph (p. 17):
“The findings in this study furthermore align with prior research showing that mo-tivational inconsistency—such as goal conflicts and avoidance goals—is linked to in-creased psychological symptoms like stress, anxiety, and depression (Fries & Grawe, 2006). This study adds experimental evidence that even situationally induced motiva-tional incongruence causes acute stress responses, suggesting a causal role. Supporting this, Fries et al. (2008) found that individuals with high goal conflict benefit more from clarification-focused therapy. Similarly, Gorges et al. (2022) demonstrated that goal conflicts undermine basic psychological needs at work, increasing stress. In addition, Franco and Trombetta (2010) found that discrepancies between employees’ desired and actual work conditions (so-called GAPs) more strongly predicted well-being than absolute perceptions of the work environment, emphasizing the importance of per-ceived congruence. Together, these results highlight motivational consistency as cru-cial for psychological well-being and suggest that addressing motivational conflicts may reduce psychological strain across populations.”
References:
Franco, M., & Trombetta, M. (2010). Cybernetic theory as a new approach to studying workers’ well-being. Croatian Economic Survey, 12(1), 27–55. URI: https://hrcak.srce.hr/52494
Gorges, J., Neumann, P., & Störtländer, J. C. (2022). Teachers between a rock and a hard place: Goal conflicts affect teaching motivation mediated by basic need satisfaction. Frontiers in Psychology, 13, 876521. https://doi.org/10.3389/fpsyg.2022.876521
Round 2
Reviewer 2 Report
Comments and Suggestions for Authors
Almost all my concerns have been put into the limitations.